# The Effect of Bisphenol A on the IVF Outcomes Depending on the Polymorphism of the Detoxification System Genes

**DOI:** 10.3390/jpm11111091

**Published:** 2021-10-26

**Authors:** Anastasiya Syrkasheva, Vladimir Frankevich, Svetlana Kindysheva, Nataliia Starodubtseva, Andrey Donnikov, Nataliya Dolgushina

**Affiliations:** 1V.I. Kulakov National Medical Research Center for Obstetrics, ART Department, Gynecology and Perinatology Ministry of Healthcare of Russian Federation, 4 Oparin Str., 117997 Moscow, Russia; 2V.I. Kulakov National Medical Research Center for Obstetrics, Department of Systems Biology in Reproduction, Gynecology and Perinatology Ministry of Healthcare of Russian Federation, 4 Oparin Str., 117997 Moscow, Russia; v_frankevich@oparina4.ru (V.F.); s_kindysheva@oparina4.ru (S.K.); 3V.I. Kulakov National Medical Research Center for Obstetrics, Laboratory of Proteomics of Human Reproduction, Gynecology and Perinatology Ministry of Healthcare of Russian Federation, 4 Oparin Str., 117997 Moscow, Russia; n_starodubtseva@oparina4.ru; 4V.I. Kulakov National Medical Research Center for Obstetrics, Department of Molecular Genetic Methods, Gynecology and Perinatology Ministry of Healthcare of Russian Federation, 4 Oparin Str., 117997 Moscow, Russia; donnikov@dna-technology.ru; 5V.I. Kulakov National Medical Research Center for Obstetrics, R&D Department, Gynecology and Perinatology Ministry of Healthcare of Russian Federation, 4 Oparin Str., 117997 Moscow, Russia; n_dolgushina@oparina4.ru

**Keywords:** assisted reproductive technologies, embryo, pregnancy, endocrine disruptors, bisphenol A, HPLC-MS, detoxification genes, gene polymorphism (SNP)

## Abstract

The aim of the study was to analyze the relationship between the level of bisphenol A (BPA) in the blood and follicular fluid, the polymorphism of the detoxification system genes, and the outcomes of IVF cycles. The data of 300 infertile patients with fresh IVF-ET cycles were analyzed. The level of BPA in the blood and follicular fluid was determined by HPLC-MRM-MS/MS. Determination of genotypes of the detoxification system genes was carried out by the real-time PCR. The threshold level for determining BPA was 0.1 ng/mL. BPA was detected in 92.3% (277/300) blood and in 16.8% (49/292) follicular fluid (FF) samples. There was no correlation between BPA level in the blood and FF. In patients with the absence of the A allele of the *SULT1A1* gene, BPA was detected in FF significantly more often (22.6% vs. 13.5%, *p* = 0.0341). There was an association (not statistically significant) between the level of BPA in the blood and the presence of the G allele of the *GSTP1* gene (rs1695) and the C allele in the *GSTP1* gene (rs1138272). Our data suggests the role of detoxification system genes in the metabolism of BPA in the human body. The influence of BPA and detoxification system genes on the IVF outcomes requires further research.

## 1. Introduction

Bisphenol A (BPA) is an industrial chemical widely used in the production of polycarbonate plastics and epoxy resins. BPA is a component used in the manufacture of disposable tableware, water bottles, cash register receipts, and other plastic products. The widespread use of bisphenol A in the food industry is of concern to the scientific community, as this substance can enter the human body. Population studies show that bisphenol A is detected in 95–100% of healthy volunteers [1,2].

Due to its structural similarity to 17β-estradiol and its ability to bind to estrogen receptors, BPA has estrogenic activity. It was found that BPA is an “endocrine disruptor”, which means the ability to negatively affect the function of the endocrine system.

The human reproductive system is vulnerable to endocrine disruptor chemicals, since its function is regulated by the action of gonadotropins and steroid hormones, which are sensitive to external factors. There is also evidence of a direct negative effect of BPA on reproductive tissues andmitotic and meiotic cell division [3,4]. Despite the large amount of studies, the pathogenetic mechanisms of BPA toxicity remain unclear.

The body’s susceptibility to environmental factors can also depend on individual characteristics. Detoxification genes, which are characterized by significant population diversity, play an important role in the biotransformation and excretion of industrial chemicals, including BPA. The role of polymorphism of the detoxification genes has been shown in studies of drug metabolism, cancer, and allergic diseases [5,6,7].

The study of the effect of BPA and the genetic characteristics of the detoxification system on reproductive health is an urgent topic of research.

The aim of the study—to analyze the relationship between the level of bisphenol A (BPA) in the blood and follicular fluid, the polymorphism of the detoxification system genes, and the outcomes of assisted reproductive technologies (ART) programs in patients with infertility.

## 2. Methods

### 2.1. Ethical Approval

All experimental protocols and methods are approved by the Ethical Committee of the National Medical Research Center for Obstetrics, Gynecology and Perinatology named after Academician V.I. Kulakov of the Ministry of Healthcare of Russian Federation, Moscow (protocol No 10, 16 October 2016). All clinical investigations are conducted according to the principles expressed in the Declaration of Helsinki. All the patients signed informed consent.

### 2.2. Patient Selection

This was a single-center prospective cohort study. Patients who were undergoing IVF with gonadotropin and a GnRH antagonist for controlled ovarian hyperstimulation (COH) were enrolled from January 2017 to December 2018.

The inclusion criteria were normal karyotype, the absence of severe male factor (100% teratozoospermia, absolute asthenozoospermia, all types of azoospermia), woman’s age from 18 to 39 years, and woman’s body mass index (BMI) from 19 to 29 kg/m^2^. The exclusion criteria were the use of donor gametes or surrogacy, and three or less oocytes on the day of oocyte retrieval.

### 2.3. Controlled Hyper-Stimulation Induction and Embryo Transfer

COH was performed using GnRH antagonist protocol. The clinician selected the appropriate gonadotropin dose for each patient on an individual basis according to the patient characteristics. Gonadotropin therapy was initiated on day 2–3 of the menstrual cycle. On stimulation day 6, a GnRH antagonist 0.25 mg/day was initiated and continued throughout the stimulation period. Triggering of final follicular maturation (human chorionic gonadotropin 10,000 IU) was performed as soon as three or more follicles were 17 mm in diameter. Oocyte retrieval took place 36 h after triggering of final follicular maturation.

Oocyte retrieval, embryological stage, and embryo transfer were performed routinely.

### 2.4. Data Collection

The collection of blood samples was carried out on the day of oocyte retrieval, after which the samples were cryopreserved at t −70 °C. Follicular fluid (FF) was collected immediately after oocyte retrieval. For this study, we selected FF samples from the first 2–3 aspirated follicles. Samples contaminated with blood were excluded from the study.

### 2.5. HPLC-MS Methods

Benzophenone (99%, Sigma-Aldrich, St. Louis, MO, USA) was used as an internal standard. Preliminary derivatization with Dansylchloride (99%, HPLC-grage, Sigma-Aldrich, St. Louis, MO, USA) was carried out to increase the sensitivity of the method. All stock solutions were prepared by dissolving the required amount of bisphenol A (4,4′-isopropyl-idenediphenol, 228.29 g/mol, Sigma-Aldrich, St. Louis, MO, USA) or benzophenone in 100% MeOH (99.9%, HPLCgrade, Scharlau, Germany). The solutions were stored in a refrigerator at a temperature of 40 °C for no more than 1 week in glass vials.

Chromatographic separation was performed on a 1260 Agilent HPLC (Agilent Technologies, Santa Clara, CA, USA) with mass spectrometric detection (QTRAP 5500 ABSciex, Vaughan, ON, Canada) in ESI-MS/MS mode in positive MRM mode. C18 Zorbax Eclipse column (3 × 50 mm, 1.8 μm, Agilent, Santa Clara, CA, USA) was used for analysis. Mobile phase A was a distilled water with 0.1% formic acid, and mobile phase B-acetonitrile with 0.1% formic acid. The measurements were carried out in an isocratic mode for 5 min with A/B = 4/96 at a flow rate of 450 μL/min. The column temperature was 35 °C, and a sample volume of 5 μL was used. HPLC-MS/MS analysis was performed in three technical replicates.

BPA measurement was performed in the blood and FF.

### 2.6. Measurement of Detoxification Gene Polymorphism

Determination of polymorphic loci of glutathione-S-transferase T1 (*GSTT1*, gene deletion), glutathione-S-transferase M1 (*GSTM1*, gene deletion), glutathione-S-transferase P1 (*GSTP1,* rs1695)), cytochrome P450 superoxide dismutase (*SOD*, rs4880), CYP1A1, (rs4646903, rs1048943, rs 1799814), glutathione peroxidase 1 (*GPX1*, rs 1050450), epoxide hydrolase 1 (*EPH171*, rs1051740), N-acetyl transferase 2 (*NAT2*, rs 1801280, rs 179993, rs 179930), and sulfotransferase 1A1 (*SULT1A1*, rs9282861) was performed by real-time polymerase chain reaction. The laboratory did not have access to the clinical characteristics of the patients. Detoxification gene measurement was performed in the blood.

### 2.7. Outcome Measures

The outcomes of IVF cycles were assessed: fertilization rate (<90% or ≥90%), blastocyst formation rate (<30% or ≥30%), clinical pregnancy rate, live birth rate, and cumulative birth rate.

### 2.8. Statistical Analysis

The statistical software package Statistica 12 (USA) was used. All continuous variables are expressed as median (interquartile range) and categorical variables as frequencies or percentages. Statistical analysis was performed using the χ2 test for comparing categorical variables and the Kruskal–Wallis/Mann–Whitney test for comparing continuous data. Correlation analysis was performed using Spearman’s test. Regression analysis was used to assess the effect of the level of bisphenol A in the blood/follicular fluid on the IVF outcomes programs depending on the detoxification genes variants. *p* value of  < 0.05 was considered significant.

## 3. Results

### 3.1. Characteristics of the Study Population

All patients included in the study were middle aged and had a normal BMI and ovarian reserve. The main characteristics of patients and IVF cycle characteristics are presented in Table 1.

### 3.2. Measurement of BPA in the Blood and Follicular Fluid

The threshold level for determining bisphenol A (both in the blood and in the follicular fluid) was 0.1 ng/mL. BPA was detected in 92.3% (277/300) of blood samples and in 16.8% (49/292) of follicular fluid (FF) samples. There was no correlation between BPA level in the blood and FF in the same patients (r = −0.1566; *p* = 0.2877). The distribution of BPA in the blood is presented on Figure 1. Most of the values were in the range from 0.1 ng/mL to 1.0 ng/mL, which is consistent with other studies [2,3]. However, we also found some outliers (maximum 56.43 ng/mL). We suppose that the cause of this phenomenon could be the consumption of a high dose of BPA before blood collection.

### 3.3. Relationship between BPA, Main Patient’s Characteristics and Detoxification Gene Polymorphism

We analyzed the main characteristics of the patients and its relationship with the level of BPA in the blood and follicular fluid. We examined the median of BPA in the blood and follicular fluid, and the proportion of patients with detected level of BPA in FF (Table 2). In patients with a higher BMI (≥25), BPA was more often detected in the follicular fluid (31.6% vs. 14.6%, *p* = 0.0122) (Figure 2), and the median BPA in the blood was lower (0.38 ng/mL vs. 0.53 ng/mL, *p* = 0.0411). No other differences were found.

It was noted that in patients with the absence of the A allele of the *SULT1A1* gene, bisphenol A was detected in follicular fluid significantly more often (22.6% vs. 13.5%, *p* = 0.0341). A relationship (not statistically significant) was also found between the level of bisphenol A in the blood and the presence of the G allele of the *GSTP1* gene (rs1695), the C allele in the *GSTP1* gene (rs1138272), the frequency of BPA detection in follicular fluid and the presence of the A allele of the *CYP1A1* gene, and the presence of the C allele of the *SOD2* gene (Table 2). The level of bisphenol A in FF was not associated with the studied variants of detoxification genes.

### 3.4. Impact of BPA and Detoxification Gene Polymorphism on the IVF Outcomes

There was no difference in IVF outcomes in quartile groups of bisphenol A: clinical pregnancy, livebirth, and cumulative livebirth rates were comparable in Q1–Q4 groups (quartile groups). Then we analyzed the relationship between polymorphism of detoxification genes and IVF outcomes. Patients with the absence of the T allele of the *CYP1A1* gene had a lower birth rate (*p* = 0.0270) and a cumulative birth rate (*p* = 0.0249), which in patients with the genotype T/T was 51.6%, in patients with the T/C genotype—38.1%, and in patients with the C/C genotype—20%.

We performed a multiple regression analysis to check the influence of each predictor, detected during the univariate analysis, on the different IVF outcomes: fertilization rate, blastocyst formation rate, clinical pregnancy rate, and livebirth rate. We did not find any statistically significant association between BPA level, detoxification gene polymorphism, and IVF outcomes. Presence of BPA in FF and A allele of *CYP1A1* gene were associated with low blastocyst formation rate, although the difference was not significant (*p* = 0.0720).

Not statistically significant differences were obtained when evaluating the combined effect of allele A of the *CYP1A1* gene (rs1799814) and the presence of BPA in follicular fluid on the blastocyst formation rate. The odds ratio for having blastocyst formation rate ˂30% in case of the presence of BPA in FF and allele A of the *CYP1A1* gene was 6.3 (95% CI 0.9–133.0).

## 4. Discussion

### 4.1. Body Fluid Concentration of BPA

Bisphenol A was detected in the blood of the majority of patients included in the study (92.3%), which is consistent with literature data [4]. BPA was also detected in 16.8% of follicular fluid samples, which confirms the hypothesis of its ability to penetrate the blood–tissue barriers.

The literature data on the association of BPA levels in blood and follicular fluid are contradictory: for example, in a study by Krotz P et al., bisphenol A and phthalates are not found in detectable concentrations in follicular fluid [8]. In another study, on the contrary, BPA was detected in most samples of the follicular fluid of patients in ART programs, but there was no association between its level and the quality of oocytes/embryos [9].

These differences in research results may be due to several factors: a variety of methods for the determination of BPA, and heterogeneity of the patients. The ability of the BPA to pass the blood–tissue barrier may also depend on the duration of its exposure, as well as on the individual characteristics of the patient.

### 4.2. Factors Affecting BPA Levels

According to our data, the being overweight was associated with the presence of BPA in the follicular fluid; however, the median BPA level in the blood was higher in patients with normal body weight. The association between BPA exposure and obesity has been studied by several research groups [10,11,12].

The most likely mechanism of BPA impact on the adipose tissue is change in the function of adipocytes. Menale C. et al. demonstrated an increase in the expression of proinflammatory cytokines and genes *FABP4* and *CD36* (involved in lipid metabolism) and a decrease in the expression of the *PCSK1* gene (involved in insulin synthesis) under the exposure of BPA on adipocytes in vitro [13]. It is possible that overweight/obesity can affect the metabolism of BPA in the human body and its ability to pass through blood–tissue barriers, but this issue requires further study.

The metabolism of BPA in mammals has been mainly studied in animal models or in human tissues models in vitro. The liver is the main site of BPA metabolism, where most parts of this substance undergo biotransformation, and subsequent excretion. Experimental studies have confirmed the role of various detoxification enzymes (phase 1 and 2) in BPA metabolism [14].

In our study, the level of BPA in the blood of patients was associated with the polymorphism of the *GST* (glutathione-S-transferase) gene, although the difference was not statistically significant. Similar data were obtained in a study by Yu Min Lee et al., which showed a combined effect of BPA level and *GST* gene polymorphism on intrauterine growth restriction risk [15].

The frequency of BPA detection in follicular fluid was in association with polymorphic variants of the *SULT1A1* genes (statistically significant differences), *SOD2*, and *CYP1A1*. These are the genes of the detoxification phase 1 and 2. We did not find studies that analyzed the frequency of determination of BPA in various biological tissues, depending on the genetic characteristics of the detoxification system.

During phase 2 of detoxification (conjugation phase), hydrophilic groups are joined to the toxic substance with the appropriate enzymes (for example, sulfotransferase transfers a sulfate group). Since then, the metabolites become less toxic, more water soluble, and ready for excretion.

Probably, BPA metabolites lose their ability to pass through the blood–tissue barriers, which may explain the relationship between polymorphism of detoxification genes and the frequency of BPA detection in follicular fluid. However, this hypothesis requires confirmation in further studies.

### 4.3. IVF Outcomes

At the last stage of the study, we assessed the combined effect of variants of detoxification genes and BPA level on the IVF outcomes. The only differences were obtained when assessing the effect of *CYP1A1* and the presence of BPA in the follicular fluid on the blastocyst formation rate (*p* = 0.0720). Previous studies have noted a negative effect of BPA on the quality of gametes and embryos, however, the pathogenetic mechanisms of this phenomenon are not well understood [15]. There was no impact of BPA levels in the blood or FF on clinical pregnancy and livebirth rates.

### 4.4. Conclusions

Our data suggests that detoxification system genes play a role in the metabolism of BPA in the human body. The influence of BPA and detoxification system genes on the IVF outcomes requires further research.

## Figures and Tables

**Figure 1 jpm-11-01091-f001:**
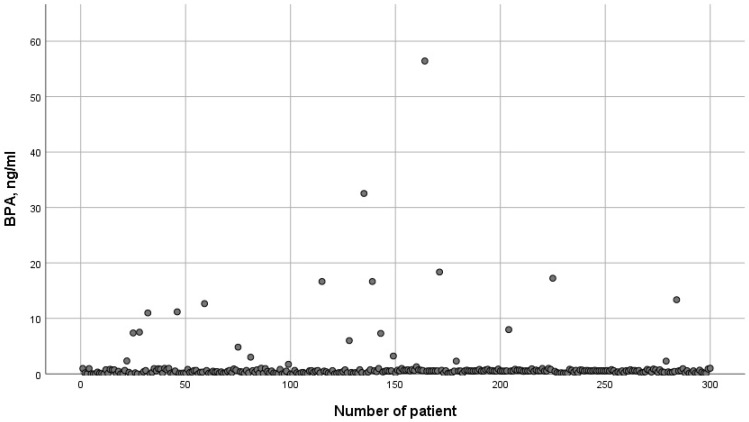
The distribution of BPA concentration in the blood.

**Figure 2 jpm-11-01091-f002:**
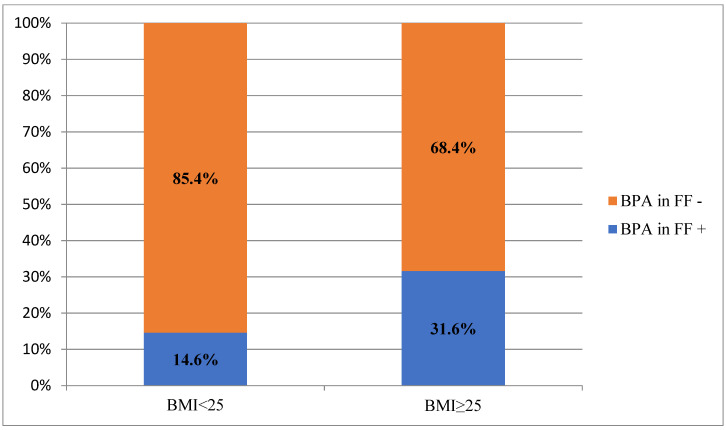
The presence of BPA in FF in accordance to female BMI.

**Table 1 jpm-11-01091-t001:** Main patients’ characteristics and IVF cycle parameters.

	Median (Interquartile Range)
Age, years	31 (29–34)
BMI, kg/m^2^	21.6 (20.4–23.5)
FSH, IU/mL	6.8 (5.1–8.2)
AMH, ng/mL	3.2 (1.9–5.8)
Antral follicle count (AFC)	12 (8–18)
Days of stimulation	9 (8–10)
Σ dose of gonadotropins	1200 (975–1500)
Oocytes retrieved	9 (5–13)
MII oocytes	7 (4–10)
Fertilization rate, %	100 (87–100)
Number of blastocysts	3 (1–4)
Blastocyst formation rate, %	46 (22–63)

BMI = body mass index; FSH = follicle stimulating hormone; AMH = Anti-Mullerian hormone.

**Table 2 jpm-11-01091-t002:** The concentration of bisphenol A in the blood and FF of patients depending on main patient’s characteristics and polymorphisms of detoxification system genes.

	BPA in Blood, ng/mL		Presence of BPA in FF, *n* (%)		BPA in FF, ng/mL	
	ME (IQR)	*p*-Level *	ME (IQR)	*p*-Level **	ME (IQR)	*p*-Level *
Age (years)		0.8810		0.2151		0.7943
<30	0.52 (0.26–0.69)		12/91 (13.2%)		0.40 (0.20–0.60)	
30–35	0.52 (0.18–0.62)		29/139 (20.9%)		0.20 (0.20–0.70)	
≥35	0.52 (0.18–0.71)		8/61 (13.1%)		0.20 (0.15–0.85)	
BMI (kg/m^2^)		0.0411		0.0122		0.8798
<25	0.53 (0.22–0.68)		37/253 (14.6%)		0.20 (0.20–0.70)	
≥25	0.38 (0.16–0.59)		12/38 (31.6%)		0.35 (0.15–0.80)	
Smoking		0.6043		0.1445		0.5520
Yes	0.48 (0.18–0.69)		6/22 (27.3%)		0.30 (0.10–0.50)	
No	0.52 (0.20–0.66)		43/269 (16.0%)		0.30 (0.20–0.70)	
AMH, ng/mL		0.5336		0.4420		0.4016
≥1.2	0.52 (0.19–0.66)		43/250 (17.2%)		0.20 (0.10–0.70)	
<1.2	0.55 (0.33–0.71)		6/41 (14.6%)		0.40 (0.20–1.10)	
*SOD2* (rs4880)		0.9355		0.0821		0.9031
C allele +	0.52 (0.20–0.65)		41/218 (18.8%)		0.30 (0.20–0.70)	
C allele −	0.52 (0.19–0.68)		8/73 (11.0%)		0.20 (0.20–0.70)	
*GSTP1* (rs1695)		0.0769		0.5121		0.1997
G allele +	0.51 (0.17–0.63)		26/152 (17.1%)		0.40 (0.20–0.80)	
G allele −	0.54 (0.25–0.72)		23/139 (16.5%)		0.20 (0.10–0.60)	
*GSTP1* (rs1138272)		0.0745		0.6050		1.0
C allele +	0.52 (0.20–0.66)		48/286 (16.8%)		0.25 (0.20–0.70)	
C allele −	0.18 (0–0.28)		1/5 (20.0%)		0.30 (0.30–0.30)	
CYP1A1 (rs1799814)		0.4865		0.0588		1.0
A allele +	0.52 (0.14–0.62)		0/15 (0%)		-	
A allele −	0.52 (0.21–0.66)		49/276 (17.8%)		0.30 (0.20–0.70)	
SULT1A1 (rs9282861)		0.3354		0.0341		0.9044
A allele +	0.52 (0.18–0.65)		25/185 (13.5%)		0.30 (0.20–0.60)	
A allele −	0.53 (0.25–0.68)		24/106 (22.6%)		0.25 (0.15–0.85)	

* Kruskal–Wallis/Mann–Whitney test ** χ^2^-test.

## Data Availability

The data presented in this study are available on request from the corresponding author.

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
