# Peer review of "The Effect of Bisphenol A on the IVF Outcomes Depending on the Polymorphism of the Detoxification System Genes"

_jpm, 2021, doi:10.3390/jpm11111091_

Round 1
Reviewer 1 Report
The manuscript: “The effect of bisphenol A on the IVF outcomes depend-ing on the polymorphism of the detoxification system genes” aimed to analyze the relationship between the level of bisphenol A (BPA) in the blood and follicular fluid, the polymorphism of the detoxification system genes and the outcomes of ART programs in patients with infertility. Some potentially interesting data were obtained. Nevertheless, to my opinion, the manuscript has serious issues in methodology, description, and interpretation of the data obtained previously and by the authors. The reader ends up doubting that authors understand the actual meaning of the results and it is difficult for the reader to get their point of view.
Specific comments
- Numerous methodological details are missing, especially when it comes to follicular fluid preparation (e.g. have you pooled FF from different follicles? If so, how many follicles have been used?). As you should know, FF aspirates contain granulosa-lutein cells and non-steroidogenic cells mainly leukocytes, to various degree, therefore it remains unknown for the reader where exactly detoxification genes were analyzed. This issue should be clarified, and details regarding RNA extraction, RT-PCR, and Real-time PCR should also be provided. Moreover, the ovarian stimulation protocol should be briefly described and the specific allele frequency in the study population should also be given.
- The introduction and discussion sections are too general, and the discussion does not include interpretation/discussion of obtained results. In addition, it would be good if each paragraph in this section ends with the concluding statement. The importance, novelty and implications (clinical relevance) of the obtained results should be emphasized at least in the last paragraph of the Discussion section – but this is missing in this manuscript. The authors should have also better justified the choice of genes tested. Without these, the biological sound of the paper is really weak, and readers do not understand the actual meaning of the results.
- The results are of great concern as to their description and interpretation, because in many places in the manuscript the authors describe the relationship between BPA and specific gene, although the difference was not statistically significant or even there was no tendency. This makes the work lose credibility.
see for instance “the level of BPA in the blood of patients was associated with the polymorphism of the GST (glutathione-S-transferase) gene, although the difference was not statistically significant” or “The frequency of BPA detection in follicular fluid was in association with polymorphic variants of the SULT1A1 genes (statistically significant differences), SOD2” – if there is no difference (P=0.08 for SOD2 is not even a tendency), how can you say there was a relationship?
Other comments:
- In my opinion, more figures or tables should be added to reflect the whole results section, to make it more readable and interesting. For example, results described in subsection 4. Impact of BPA and detoxification gene polymorphism on the IVF outcomes could be presented as table.
- Please explain what are Q1-Q4 groups?
- I have not heard the term PCR-RT (see the abstract) .........maybe Real-time RT-PCR?
- Table 1, I suggest to delete (*), it’s unnecessary here
Author Response
Dear reviewer!
Please, accept my sincere appreciation for your help with our article.
- We pooled FF from different follicles. For this study we selected FF samples from the first 2-3 follicles and excluded samples contaminated with blood. Detoxification genes were analyzed in the blood, not in the FF.
On page 3 following text was added: “For this study we selected FF samples from the first 2-3 aspirated follicles. BPA measurement was performed in the blood and FF. Detoxification gene measurement was performed in the blood.”
Ovarian stimulation protocol description added to MM, page 2: Gonadotropin therapy was initiated on day 2–3 of the menstrual cycle. On stimulation day 6, a GnRH antagonist 0.25 mg/d was initiated and continued throughout the stimulation period. Triggering of final follicular maturation (human chorionic gonadotropin 10 000 IU) was performed as soon as three or more follicles were 17 mm in diameter. Oocyte retrieval took place 36 hours after triggering of final follicular maturation.
In my opinion, it is better to present allele frequency of studied genes in Supplement, not in the main text. That table was added in Supplement file.
- The choice of detoxification genes was based on scientific papers. Several studies demonstrate the role of detoxification genes in BPA and BPS metabolism in animal studies (doi: 10.1186/s12864-020-07294-3). Other studies demonstrate the role of those genes in industrial chemicals excretion and drug metabolism (doi: 10.1007/s11033-019-05143-5, doi: 10.2174/1389450118666170125144557).
The Discussion was improved. The paragraphs were added to make the article easier to read.
- We found statistically significant difference (p˂0.05) for some genes and presence of BPA in FF, for some genes p-value was between 0.05 and 0.09. According to special aspects to this study (measurement of endocrine disruptors and detoxification gene polymorphism in IVF patients, rare topic in reproduction), we considered p-value less than 0.09 as a tendency.
Other comments:
- In my opinion, it is better to present the table with IVF outcomes in Supplement (added to Supplement).
- Q1-Q4 mean quartile groups. Clarification of this term was added to article, page 7.
- Real time PCR: added to abstact.
- Table 1 was improved
Reviewer 2 Report
I read with great interest the manuscript, which falls within the aim of this Journal. In my honest opinion, the topic is interesting enough to attract the readers’ attention. Nevertheless, authors should clarify some points and improve the discussion, as suggested below.
Authors should consider the following recommendations:
- Manuscript should be further revised in order to correct some typos and improve style.
- I recommend to add further detail about the correlation between exposure to environmental toxicants and endometriosis, since the latter is clearly associated with infertility in 50% of the cases (authors may refer to: PMID: 25920525).
- To date many efforts are spent to identify a correct algorithm which considers woman's age and ovarian reserve markers as a tool to optimize the recombinant follicle-stimulating hormone (rFSH) starting dose in IVF procedure. Nevertheless, current available evidence regarding PCOS women, particularly the ones with high AMH, does not seem adequate. I would be glad if the authors discuss this important point, referring to: PMID: 30242498.
Author Response
Dear reviewer!
I truly appreciate your kindness for reviewing our study.
- The relationship between environmental toxicants and endometriosis is a relevant topic for future studies, but in this study severe endometriosis was an exclusion criterion. The mild endometriosis was not an exclusion criterion, but there was no any relationship between endometriosis and BPA level/gene polymorphism/IVF outcomes.
- Normal ovarian response (4-15 oocytes) was an inclusion criterion, we excluded patients with PCOS, poor/hyper ovarian response.
Round 2
Reviewer 1 Report
The manuscript has been sufficiently improved and, in my opinion, it can be accepted for publishing.